# Exploring the Potential of Olive Flounder Processing By-Products as a Source of Functional Ingredients for Muscle Enhancement

**DOI:** 10.3390/antiox12091755

**Published:** 2023-09-13

**Authors:** Jimin Hyun, Sang-In Kang, Sang-Woon Lee, R. P. G. S. K. Amarasiri, D. P. Nagahawatta, Yujin Roh, Lei Wang, Bomi Ryu, You-Jin Jeon

**Affiliations:** 1Department of Marine Life Sciences, Jeju National University, Jeju 63243, Republic of Korea; localman@office.jejunu.ac.kr (J.H.);; 2Seafood Research Center, Silla University, Busan 49277, Republic of Korea; ftrnd5@silla.ac.kr; 3College of Food Science and Engineering, Ocean University of China, Qingdao 266003, China; 4Department of Food Science and Nutrition, Pukyong National University, Busan 48513, Republic of Korea

**Keywords:** fish by-product, dexamethasone-induced muscle atrophy, dexamethasone-induced ROS production, myogenesis, enzymatic hydrolysis, zebrafish

## Abstract

Olive flounder (OF) is a widely aqua-cultivated and recognized socioeconomic resource in Korea. However, more than 50% of by-products are generated when processing one OF, and there is no proper way to utilize them. With rising awareness and interest in eco-friendly bio-materialization recycling, this research investigates the potential of enzymatic hydrolysis of OF by-products (OFB) to produce functional ingredients. Various enzymatic hydrolysates of OFB (OFBEs) were generated using 11 commercial enzymes. Among them, Prozyme 2000P-assisted OFBE (OFBP) exhibited the highest protein content and yield, as well as low molecularization. The muscle regenerative potential of OFBEs was evaluated using C2C12 myoblasts, revealing that OFBP positively regulated myoblast differentiation. In an in vitro Dex-induced myotube atrophy model, OFBP protected against muscle atrophy and restored myotube differentiation and Dex-induced reactive oxygen species (ROS) production. Furthermore, zebrafish treated with OFBEs showed improved locomotor activity and body weight, with OFBP exhibiting outstanding restoration in the Dex-induced muscle atrophy zebrafish in vivo model. In conclusion, OFBEs, particularly OFBP, produce hydrolysates with enhanced physiological usability and muscle regenerative potential. Further research on its industrial application and mechanistic insights is needed to realize its potential as a high-quality protein food ingredient derived from OF processing by-products.

## 1. Introduction

Fish are an important part of diets worldwide, providing essential nutrients such as functional proteins with amino acids (AAs), omega-3 fatty acids, vitamin D, and essential minerals with a unique taste [1]. Although fish preferences may vary depending on culture and geography, incorporating fish into the general diet can be an effective way to improve overall health conditions. Due to advancing aqua-cultivation skills and proximity to an extensive coastline, South Korea is a leading consumer of fish in varied forms, including raw and cooked [2,3].

South Korea is a leading global producer of *Paralichthys olivaceus*, known as olive flounder (OF), and OF aquaculture is an important domestic industry. According to the Ministry of Oceans and Fisheries of South Korea, OF aquaculture production was approximately 43,813 tons in 2020. OF is an important fish species, accounting for 65% of marine-farmed fish consumption and 50% of fish production in South Korea [4].

Recently, because of COVID-19 and an increase in single-person households as well as the elderly population, processed raw fish products that can be eaten quickly and easily at home or outside are being increasingly preferred over eating at existing restaurants. However, once edible raw fish meat separated from OF by-products (OFB), such as the head, gut, skin, and bones, accounts for approximately 50% of the total processing volume, increasing the burden of processing costs [5]. Despite being a high-quality nutrient source containing abundant PUFA, minerals, and proteins like collagen, OFB has been treated as a waste resource; however, research on its specific utilization remains insufficient [5].

By changing the original properties of the protein, protein hydrolysates obtained by protease hydrolysis of animal materials can acquire new functions [6]. This is because peptidation or free AA conversion of proteins can improve the taste and enhance the absorption rate of beneficial substances via low molecularization [7]. Such interventions have many benefits for preventing age-related diseases such as sarcopenia. Sarcopenia is characterized by muscle loss and is often observed in older individuals [8]. It is associated with geriatric diseases and can decrease the quality of life, necessitating external assistance for daily activities [8]. The economic burden of sarcopenia is estimated to be approximately USD 40 billion in the US [9]. Although aging is inevitable, medication, exercise, and nutritional supplements can improve muscle strength and physical activities in individuals with sarcopenia [10].

In the elderly, the protein synthesis ratio is lower than the protein breakdown ratio; therefore, more protein must be consumed to achieve the same level of protein synthesis as in younger people (anabolic resistance: breakdown > synthesis) [7]. However, consuming nutrients that satisfy the muscle full effect and leucine triggers is necessary to induce the expected level of muscle synthesis under these conditions [11]. The muscle full effect is a phenomenon in which postprandial muscle protein synthesis (MPS) occurs when a meal is consumed after exercise [12]. MPS then returns to basal levels within 1–2 h after eating; thus, providing high-quality AAs that are easily absorbed is critical for this process [12]. Further, the efficiency of MPS decreases when the blood concentration of leucine falls below a certain level (a leucine trigger) [12]. Therefore, the use of protease-hydrolyzed fish byproducts can be an efficient means to overcome these hurdles and induce sufficient MPS.

Therefore, this study aimed to evaluate the various aspects of enzymatic hydrolysis, including extraction yield, protein content, and promotion of myocyte differentiation, in low-molecular-weight OFB enzymatic hydrolysates, as well as to identify the optimal enzyme and assess its potential for application in large-scale production. The effect of the hydrolysate was further examined on myocyte differentiation in vitro and in a zebrafish model of dexamethasone-induced muscle atrophy.

## 2. Materials and Methods

### 2.1. Materials and Reagents

OFB sourced from local aquacultured OF, excluding gut lesions, were stored at −80 °C for the experiment and their nutritional components are described (Table 1). OFB was kindly provided by FIssueH Corp. (FissueH Corp., Jeju-do, Republic of Korea). Dulbecco’s modified Eagle’s medium (DMEM), fetal bovine serum (FBS), horse serum (HS), 100X penicillin/streptomycin (P/S), trypsin-EDTA, and phosphate-buffered saline (PBS) were provided by Gibco-BRL (Burlington, ON, Canada). The BCA assay kit for protein quantification and ECL detection reagent were purchased from Bio-Rad (Richmond, CA, USA) and Amersham Biosciences (Piscataway, NJ, USA), respectively. Antibodies against phospho-Akt (S463), AKT, phosphor-p70S6K, p70S6K, phosphor-4E-BP1, and 4E-BP1 were obtained from Cell Signaling Technology (Danvers, MA, USA). Antibodies against Atrogin-1/MAFbx, MuRF-1, phosphor-mTOR (S2448), mTOR, MyH, Myogenin, MyoD, GAPDH, and β-Actin were purchased from Santa Cruz Biotechnology (Santa Cruz Biotechnology, Dallas, TX, USA). For enzymatic hydrolysis, 11 enzymes were purchased from Novozymes Ltd. (Bagsvaerd, Denmark) and BISION-Biochem (BISION-Biochem, Gyeonggi-do, Republic of Korea) (Table 2). Further, 2-2-[N-(7-Nitrobenz-2-oxa-1,3-diazol-4-yl)amino]-2-deoxy-D-glucose (2-NBDG), 3-(4,5-Dimethyl-2-thiazolyl)-2,5-diphenyl-2H-tetrazolium bromide (MTT), dexamethasone (Dex), and all other chemicals were purchased from Sigma-Aldrich (St. Louis, MO, USA).

### 2.2. Sample Preparation

For lab-scale production, the frozen OFB was thawed at 4 °C over 20 h and was then soaked in running tap water for 1 h for defrosting. Prior to the hydrolysis of the 10% OFB substrate in distilled water with 11 kinds of enzymes added for 4–24 h under optimal conditions (1% of the enzyme against the substrate, pH 6.80–7.50), the moisture content of the thawed OFB was measured to determine its dry mass (73.53 ± 1.03%). At the end of enzyme-assisted OFB hydrolysis, the products were heat-inactivated at 100 °C for 10 min. The hydrolysates were then filtered through a filter paper with 25 µm pores (Whatman No.4, Brentford, UK) and the filtrates were evaporated using a rotary evaporator at 30 °C (N-1300, EYELA, Tokyo, Japan). The concentrates were frozen at −80 °C stored at room temperature following the completion of freeze drying (Figure 1).

### 2.3. General Composition Analysis of OFB

The proximate compositions of the OFB and its derivatives, including carbohydrates, proteins, lipids, and ash, were analyzed using the analytical methods of the Association of Official Analytical Chemists (AOAC). For analysis, raw OFB was determined using the above-mentioned general compositions based on dry mass by eliminating moisture content, and freeze-dried and spray-dried samples were reconstituted in phosphate-buffered saline (PBS) at a concentration of 10 mg/mL for estimation. The polysaccharide, protein, lipid, and ash contents were determined using standard methods [13,14,15,16].

### 2.4. Cell Culture

For in vitro screening, murine C2C12 myoblasts were purchased from the American Type Culture Collection (ATCC, Manassas, VA, USA) and cultured in DMEM containing 10% FBS with 1% P/S at 37 °C with 5% CO_2_. For myotube differentiation, cells that reached confluence beginning from 1 × 10^5^ cells/well at 48-well plate seeding were switched to DMEM containing 2% HS and 1% P/S; the medium was changed every two days up to the fourth day. Subsequently, the cells were treated with the desired doses of samples mixed with DMEM containing 2% HS, with or without the synthetic glucocorticoid drug, Dex, for two days to determine the potent effect of the samples by immunoblotting. For immunofluorescence staining, every two days starting from the initial differentiation, the indicated doses of samples with or without Dex were treated for six days. Thereafter, cells were harvested for the experiments described below.

### 2.5. MTT Assay, Glucose Uptake, and Cell Proliferation Measurement

The MTT assay was carried out in cells incubated with samples for 24 h to evaluate cytotoxicity based on mitochondrial performance by measuring absorbance at 570 nm [17]. Glucose uptake was estimated using the 2-NBDG assay, in which cells pretreated with the desired concentrations for 24 h were switched to serum-free DMEM containing 1% P/S for 1 h. A 10 mM stock solution of 2-NBDG was prepared by dissolving it in sterile phosphate-buffered saline (PBS). After 1 h, the serum-free medium was discarded, and the cells were gently washed with warm PBS to remove any residual glucose. Next, 50 µM of 2-NBDG working solution was added to the cells and incubated at 37 °C for 1 h to allow glucose uptake. After the incubation period, the 2-NBDG solution was carefully removed, and the cells were washed twice with cold PBS to stop further glucose uptake and remove excess 2-NBDG. The fluorescent glucose analogue-labelled cells were resuspended in 0.01% Triton X-100 detergent and the fluorescence intensity was determined at 475/550 nm (Ex/Em) using a Synergy HTX microplate reader (Agilent Technologies, Santa Clara, CA, USA). Cell proliferation after the sample treatment was determined using the CCK-8 colorimetric assay kit (Dojindo Molecular Technologies, Kamimashiki-gun, Kumamoto, Japan). Briefly, cells were treated with a series of sample doses during 48 h incubation and a 10% volume of CCK-8 was directly added to the cell culture media for 1 h. Subsequently, the supernatants were transferred to a clear-bottomed 96-well plate and absorbance was measured at 450 nm. All colorimetric assays were performed using a Synergy HTX^®^ system (Agilent).

### 2.6. ROS Production in C2C12 Myotube

The ROS production was determined in a fully differentiated C2C12 myotube using H_2_DCF-DA (Invitrogen, Carlsbad, CA, USA). The fully differentiated myotubes were treated with OFBP in a concentration-dependent manner for 1 h. After that, 10 μM of Dex of 10 mM of AAPH (alkyl radical) were applied for 6 h or 24 h, respectively, and 5 μM of H_2_DCF-DA was sequentially applied for 1 h. Thereafter, the fluorescence intensity was measured at 480/530 (Ex/Em) using the Synergy HTX^®^ system (Agilent).

### 2.7. Immunoblotting

C2C12 cells were treated with the indicated concentrations of OFBP for 48 h in the presence or absence of Dex (10 µM). The cells were washed with chilled PBS and lysed using radioimmunoprecipitation assay (RIPA) buffer (Sigma-Aldrich). The BCA assay (Bio-Rad, Hercules, CA, USA) was used for protein quantification in tissue lysates, which were then separated using 10% SDS-PAGE. Electroporated proteins were transferred to nitrocellulose membranes (Amersham, Little Chalfont, Buckinghamshire, UK), which were blocked with 5% bovine serum albumin (BSA; Sigma-Aldrich). The membranes were incubated overnight with primary antibodies at 4 °C. An immunoblot Chemiluminescence Reagent was used to detect the proteins using the FUSION SOLO system (Vilber Lourmat, Paris, France).

### 2.8. Immunofluorescence Staining

C2C12 myotubes were differentiated in clear-bottomed 48-well plates; following treatment, the cells were fixed with 4% paraformaldehyde for 1 h and permeabilized using 100% chilled methanol for 15 s. The cells were then washed thrice with 1X PBS + 0.1% Tween20 (PBST) every five mins. Blocking was performed with 1% BSA containing 2.252% glycine for 30 min, followed by incubation with the MyH primary antibody (1:500 in 1% BSA) for 1 h at room temperature. PBST wash was repeated three times every five min. The Alexa Fluor^®^ 647 conjugated secondary antibody (1:1000) was prepared in 1% BSA in PBST without glycine, and incubated for 1 h. PBST was repeated thrice, every five min. Nuclear staining was performed using 4′,6-diamidino-2-phenylindole (DAPI, 300 nM in PBST) for 3 min followed by three PBST washes every three minutes. Finally, 1 mL of PBS was added, and the cells were examined under a microscope. The myofibril diameter, myotube coverage, branching points, and total number of nuclei were determined using a Myotube Analyzer [18]. All images were captured using an automated LionHeart microscope (Agilent).

### 2.9. Zebrafish Experiments

Adult zebrafish were purchased from a commercial dealer (World fish aquarium Corp., Jeju-do, Korea) and housed 15 zebrafish in a 3.5 L clear acrylic tank. The tank conditions were as follows: 28.5 ± 1 °C, 14/10 h light/dark cycle, and twice-daily feeding (Tetra GmgH D-49304 Melle Made in Germany). For the Dex-induced muscle atrophy model in zebrafish, the zebrafish with constant group body weight (12.80 ± 0.80 g) were distributed to four groups: (1) Control, (2) Dex-induced sole blank, (3) DW (non-enzyme), (4) Protamex-, (5) Flavourzyme-, (6) Protana prime-, (7) Neutrase-, (8) Alcalase-, (9) Prozyme EXP 5000-, (10) Prozyme 2000P-, (11) Prozyme 1000L-, (12) Foodpro PNL-, (13) Sumizyme-assisted OFBEs (1%), and (14) Maca (3%). All groups were fed a recommended exclusive diet for zebrafish containing the indicated samples or absent for blank and control. The periods of Dex induced were 10 days and samples containing diet were begun to feed three days earlier, prior to Dex exposure. For the determination of zebrafish’s muscle atrophy release, their behaviors related to swimming, such as velocity, acceleration, total distance moved, and activity rate, were monitored and visualized using Ethovision XT software (Noldus Information Technology, Wageningen, The Netherlands).

### 2.10. Statistical Analysis

Different statistical tests were employed based on the data distribution to compare the differences in continuous variables between the two groups. An independent *t*-test was used when the data followed a normal distribution. To assess the differences in changes over time between the two groups, a paired *t*-test was applied to normally distributed data. Means across the study groups were compared using analysis of variance (ANOVA). In case of statistically significant heterogeneity between groups observed using ANOVA, a post hoc Tukey’s test was employed. Statistical significance was set at * *p* = 0.05, ** *p* = 0.01, *** *p* = 0.001, and **** *p* = 0.0001. Statistical analyses were performed using GraphPad Prism 9.2.0 (GraphPad Software, San Diego, CA, USA). Data are displayed as mean ± SD.

## 3. Results

### 3.1. Screening of Nutritional Components and the Features of Enzymatic Hydrolysis Application in OFB

OFB, derived from OF, the most widely aquacultured fish species in Korea, is known for its outstanding taste and nutritional value. The results of this analysis revealed that OFB is a highly nutritious food, primarily because of its superior protein (56.51 ± 1.16%, *w*/*w*), lipid (20.37 ± 0.32%, *w*/*w*), and ash (18.13 ± 1.65%, *w*/*w*) contents (Table 1). Based on these results, OFB was subjected to enzymatic hydrolysis to assess protein nutrition. Twelve extracts were obtained, including 11 enzymatic hydrolysates of OFB (OFBEs) generated using 11 commercial enzymes, and a single extract obtained using distilled water (Table 2). These extracts were screened to determine the major components, such as carbohydrates, proteins, lipids, and ash (Table 3). The analysis of different extract types revealed that OFBEs exhibited significant protein and lipid levels. In particular, most OFBEs exhibited protein contents exceeding 50.00% (*w*/*w*), whereas the lipid content closely followed in terms of abundance across all OFBEs (Table 3). The protein content analysis revealed that OFB_Control had the highest protein content but the lowest yield (Table 3 and Figure 2A). Although OFB_Control illustrated the highest protein level among OFBEs, OFB_Prozyme 2000P (OFBP) was confirmed to possess a high protein content (85.72 ± 6.62%, *w*/*w*) with a minor lipid content (6.38 ± 4.99%, *w*/*w*) as well as the highest yield (Table 3 and Figure 2A). The recovery rates of substances representing the yield level showed a significant value in OFBP (65.10 ± 1.90%, *w*/*w*) compared to OFB_Control (19.87 ± 0.70%, *w*/*w*) (Figure 2A). Furthermore, OFBP showed excellent hydrolysis efficiency and turbidity compared to that in the OFB_Control and other OFBEs (Figure 2C,D). Therefore, enzymatic hydrolysis of OFB is an efficient extraction method that exhibits higher bioavailability than non-enzymatic hydrolysis (Figure 1 and Figure 2). These results indicate the potential of OFB to produce hydrolysates with enhanced physiological usability.

### 3.2. Determination of the Low-Molecularization Rate OFBEs

To confirm whether enzymatic hydrolysis improved the dramatic low molecularization compared to the non-enzyme-treated group, SDS-PAGE with Coomassie blue staining was performed for 12 OFBE species (Figure 3A). The SDS-PAGE staining results revealed that the non-enzyme-hydrolyzed control (DW) displayed a wide range of molecular weight distributions in the DW lane (Figure 3A). In contrast, most of the OFBEs had marked molecular weight distributions of less than 10 kDa (Figure 3A). To precisely estimate the low molecularization rate of OFBEs, a major list of OFBEs was determined based on HPLC-MS analysis (Figure 3B). In line with the SDS-PAGE results, the non-enzymatically hydrolyzed group comprised substances with higher average molecular weights (949.33 Da) (Figure 3B). Intriguingly, among the other types of OFBEs, Protamex—(438.67 Da), Protana prime—(426.20 Da), Prozyme EXP 5000—(433.70 Da), Sumizyme-assisted OFBE (412.53 Da), and OFBP (456.77 Da) demonstrated that more than 70% of the molecules were assigned to less than 500 Da (Figure 3B). Taken together, OFBP showed a remarkable extraction yield and hydrolysis efficiency, with a notably low molecularization rate compared to that of the other OFBEs.

### 3.3. Evaluation of the Muscle Regenerative Potential of OFBEs Using C2C12 Myoblasts

The muscle regeneration potential of OFBEs was investigated using C2C12 myoblasts. Initially, the OFBEs were tested for cytotoxicity at five concentrations (50, 100, 200, 400, and 800 µg/mL) in fully differentiated C2C12 myotubes. None of the OFBE concentrations showed any toxic effects, except for Protana prime-assisted OFBE at higher doses (Figure 4A). Based on these results, a single dose of OFBEs (300 µg/mL) was determined to regulate myotube generation via glucose uptake activity in fully differentiated C2C12 myotube cells (Figure 4B). Most OFBEs treatments markedly upregulated glucose uptake compared to that in the control group; these results were not observed in the non-enzymatically hydrolyzed group (Figure 4B). Further, OFBEs distinctly enhanced the cell proliferation ability in myoblasts in a dose-dependent manner (Figure 4C). Furthermore, the remarkable increase in cell proliferation capacity induced by OFBEs was not inferior to that in the positive control group, Maca (Figure 4C). Thus, the low-molecularized OFBEs can assist in glucose metabolism and proliferation of C2C12 myocytes in vitro.

To determine whether OFBE supplementation could boost normal myotube differentiation in C2C12 myoblasts, the expression of myogenic regulatory factors (MRFs), including MyH, MyoD, Myogenin, and MuRF-1, was investigated by immunoblotting in fully differentiated C2C12 myotubes incubated with OFBEs (Figure 4D,E). Treatment with most OFBEs, depending on the concentration, significantly regulated the protein expression of MRFs, and the improvement in MyH protein expression provided partial insight into how OFBE treatment in the Dex-induced myotube atrophy model resulted in an improvement (Figure 4D,E). Flavourzyme-assisted OFBE and OFBP treatment increased the expression of these differentiation-inducing factors, even at low concentrations, and were found to have a more significant impact without regulating the expression of the protein-degrading enzyme MuRF-1 (Figure 4D,E). Based on a comprehensive analysis of previous experimental results, OFBP exhibited the highest protein content and yield, with a high hydrolysis rate (Table 3 and Figure 2). Considering its low molecular weight and ability to promote myoblast differentiation, OFBP was selected as the most suitable candidate for further experiments.

### 3.4. The Protective Effect of OFBEs in Dex-Induced Myotube Atrophy in C2C12 Cells

C2C12 myoblasts were differentiated into myotubes to investigate the protective activity of OFBE supplementation against Dex-induced myotube atrophy in vitro (Figure 5). The tripled concentration range of OFBEs was added along with 10 μM of Dex stimulator. Myosin heavy chain (MyH)-targeted immunofluorescence staining showed that treatment with Dex alone (blank) obviously disturbed normal myotube differentiation, as determined by myotube diameters compared to the control group, and this defect was clearly recovered upon maca treatment along with Dex during myoblast differentiation (Figure 5A,B). As expected, OFBEs significantly inhibited the Dex-mediated disruption of myotube formation compared to that in the blank group (Figure 5A,B). In particular, Flavourzyme-, Protana prime-, Alcalase-, Prozyme 1000L-, and Sumizyme-assisted OFBEs, including OFBP, significantly restored Dex-induced myotube atrophy in a dose-dependent manner, and the OFBP-treated group showed great recovery, even at the lowest concentrations, similar to that in the maca group (Figure 5A,B). Consequently, OFBE supplementation protected against Dex-induced myotube atrophy in vitro, restoring normal myotube differentiation with their diameters, particularly with Flavourzyme-, Protana prime-, Alcalase-, Prozyme 1000L-, and Sumizyme-assisted OFBEs, including OFBP.

### 3.5. Restoration of Dex-Induced Muscle Atrophied Zebrafish by Treatment with OFBEs

To investigate whether OFBEs could attenuate Dex-induced muscle atrophy and promote recovery, a zebrafish model was used to examine whether OFBEs could restore muscle atrophy in response to Dex stimulation in vivo (Figure 6A). Treatment with Dex resulted in significant locomotor impairment, as evaluated by various factors, including velocity, acceleration, and total distance moved in the blank group (Figure 6B–E). Further, Dex stimulation markedly lowered body weight compared to that in the control group (Figure 6F).

Interestingly, supplementation with a 3% Maca-containing fish diet significantly reinforced swimming locomotion, as indicated by the mentioned indicators of muscle strength in zebrafish: velocity acceleration, total distance moved, and relatively improved body weight compared to that in the blank group weakened by Dex treatment (Figure 6B–F). Along with maca, 1% OFBEs-containing fish diets applied to zebrafish for a week were also assessed for locomotion tracking to determine their positive regulation of muscle atrophy (Figure 6). In particular, diets supplemented with 1% Flavourzyme-OFBE and OFBP showed outstanding improvements in locomotor activity, which served as an indicator of muscle strength in zebrafish (Figure 6B–E). Intriguingly, OFBP-fed zebrafish showed a striking restoration of body weight compared with that in the blank group stimulated with Dex alone (Figure 6F). However, several OFBEs-supplemented zebrafish groups, including a distilled water-assisted OFB hydrolysate (DW), Protana prime-assisted OFB hydrolysate (Protana-prime), Alcalase-assisted OFB hydrolysate (Alcalase), and Prozyme EXP 5000-assisted OFB hydrolysate (Prozyme EXP 5000), were not able to protect the significant decrease in their bodyweight by Dex induction in comparison to the Blank zebrafish group (Figure 6F). This confirms that OFBP consistently exhibited improvement across different experimental measures, reinforcing its positive impact on muscle function in the zebrafish model of Dex-stimulated muscle atrophy.

### 3.6. Improvement of C2C12 Myoblast Differentiation by OFBP

To verify the positive regulatory effect of OFBP on muscle atrophy, OFBP was tested in a dose-dependent manner via incubation during the differentiation of C2C12 myoblasts. Treatment with a tripled range of OFBP concentrations (30, 100, and 300 μg/mL) dramatically increased the myotube diameter (Figure 7A,B). Moreover, OFBP increased the percentage of myotube coverage, an index that describes the number of myotubes on the indicated surface (Figure 7C). The potent muscle-generative effect of OFBP was revealed by an increase in the protein expression of the myoblast myogenic regulatory factors (MRFs) MyH, MyoD, and Myogenin (Figure 7D). As shown in Figure 7A,B, treatment with a series of OFBP concentrations resulted in outstanding recovery in myofibril diameters and the percentage of myotube coverage against Dex-stimulated myotube atrophy in a dose-dependent manner (Figure 7E–G). Dex treatment suppressed the expression of MyH and Myogenin, whereas OFBP restored this damage in a concentration-dependent manner to the original protein expression levels (Figure 7H). In addition, a dramatic reduction in ROS production was intervened by the application of OFBP in C2C12 myotubes, which was stimulated by Dex (Figure 7I). Interestingly, a series of OFBP treatments with different doses showed outstanding ROS scavenging effects irradiated by AAPH (alkyl radical) in C2C12 myotubes (Figure 7J). Therefore, the remarkable concentration-dependent improvement in myoblast differentiation and alleviation of Dex-induced myotube atrophy and ROS production clearly demonstrated the functional efficacy of OFBP.

## 4. Discussion

Fish are an important component of diet worldwide because they greatly improve overall health with their rich nutritional profiles. With widespread recognition of the preventive effects of fish consumption on diseases such as muscle wasting-induced sarcopenia through large-scale cohort studies, fish-based diets are gradually gaining popularity [19,20,21]. However, as fish consumption increases, appropriate utilization of fish processing by-products remains limited [22]. This not only poses challenges for individuals and businesses but also contributes to the growing environmental burden. By-products of fish processing include residues generated during the procedure, including bones, skin, organs, and other parts [5]. However, these is a rich source of proteins and other nutrients similar to the mainly consumed portion; however, these by-products are often discarded or used as animal feed [5]. Therefore, attempts to enter the global functional food market by developing functional ingredients derived from fish are underway [23,24]. However, the development of ingredients using fish by-products has been limited because of difficulties in quality control and the incorrect stigma of being inedible. Therefore, the current research focuses on OFBE production and suggests its potential against muscle-wasting conditions by means of in vitro and in vivo model screening to propose the potential utilization of fish by-products as promising ingredients in the functional food industry for developing health-enhancing products.

Once the hydrolysis efficiency, yield rate, general composition profile, and low molecularization were evaluated in all OFBEs, OFBP was found as an outstanding candidate originating from OFB using Prozyme 2000P, an exopeptidase that can cleave AAs from the ends of polypeptide chains (Table 2 and Figure 3). Exopeptidase offers several benefits. First, it demonstrates high efficiency in complete hydrolysis by breaking down proteins and generating free AAs or peptides, thereby enhancing the product flavor [7]. Secondly, it promotes the reduction of the molecular size of the hydrolysate, which can improve the product bioavailability [6]. Lastly, exopeptidase exhibits greater stability than endo-type proteases because of its smaller active site, resulting in a lower possibility of activity inhibition and ensuring consistent product quality [25]. Thus, the advantages of using Prozyme 2000P exopeptidase for OFBP production have excellent potential for industrial applications.

Aging leads to muscle loss and decreased muscle strength; however, a sufficient intake of high-quality proteins can minimize muscle protein breakdown and promote muscle synthesis. Therefore, highly proteinaceous OFBEs along with OFBP were investigated for their muscle regenerative potential using C2C12 murine myoblasts (Figure 4D,E and Figure 7). MyoD and Myogenin are prominent MRFs that respond to dietary protein supplementation in conjunction with exercise treatment, and their expression is crucial for muscle formation [26,27]. All the mechanisms involved in muscle formation are inhibited when their expression is genetically or chemically hindered [28]. It is well known that overexpression of MyoD, even in unrelated tissue-derived cells, can regulate the fate of these cells towards muscle cell differentiation [29]. In the current study, we selected a maca containing triterpene, which is known to cause muscle hypertrophy as a positive control [30,31]. The active substances in maca have been reported to stimulate the Akt-mTOR pathway that is in charge of muscle hypertrophy by their phosphorylation and inhibit proteolysis enzymes in vitro [31]. In line with that, maca treatment not only displayed dramatic upregulation in MRFs and myogenesis (Figure 4 and Figure 7) but also prevented Dex-induced myogenic fate disturbance (Figure 5, Figure 6 and Figure 7). Intriguingly, most OFBEs significantly upregulated the protein expression of these MRFs; however, OFBP was the only one that controlled the MRFs expression in a dose-dependent manner (Figure 4D,E). These results suggest that treatment with OFBP distinctly leads to a myogenic fate in C2C12 murine myoblasts, which is attributed to the exceptional protein content and low molecular weight of OFBP, contributing to its predictably superior biocompatibility.

Dex is a common synthetic glucocorticoid that can induce muscle wasting in patients after prolonged application at high doses [32]. The high-dose Dex-induced muscle atrophy model disrupts the balance between protein synthesis and degradation, leading to muscle fiber damage and atrophy [33]. This model is widely utilized, as it allows the easy and efficient reproduction of naturally occurring muscle wasting conditions, regardless of in vitro or in vivo models [34]. Dex treatment in zebrafish has been reported to significantly induce muscular atrophy, leading to decreased swimming endurance and impaired motor performance [35]. The current results support previous reports that this model effectively replicates natural muscle atrophy and is highly suitable for studying this disease (Figure 6B). Improvement in Dex-induced muscle atrophy at the in vitro level was evident in the treatment groups using Flavourzyme-, Protana prime-, Alcalase-, Prozyme 1000L-, and Sumizyme-assisted OFBE, including OFBP. However, the administration of these samples, except OFBP, did not lead to distinct improvements in the zebrafish model of Dex-induced behavioral immobility and bodyweight reduction (Figure 6B–F). In addition, there are several points to consider as to why, unlike OFBP, which showed significance in bodyweight change, the other four sample-treated groups showed a decreasing pattern. Excluding the DW group, the average molecular weights of the remaining three groups (Protana prime, Alcalase, and Prozyme EXP 5000) were 426.20, 521.71, and 433.70 Da, respectively, indicating that these hydrolysates were high in low-molecularization (Figure 3B). This suggests that in addition to low molecularization, there are additional factors that protect Dex-induced zebrafish from muscle atrophy and inhibit weight loss. Recently, the content of leucine among the constituent amino acids has been reported to be important as a factor inducing muscle synthesis in vivo [36]. In order for muscle synthesis to occur in vivo, if a certain amount of leucine is not present in the bloodstream within one hour after a meal, muscle synthesis does not occur, and this phenomenon is called the leucine trigger [37]. Therefore, in subsequent studies, we intend to compare the differences between each enzymatic hydrolysate via the analysis of constituent amino acids and predict their potential for industrialization.

Interestingly, we observed a reduction in swimming activity and lingering at the bottom of the tank in the Dex-induced zebrafish muscle atrophy model (Figure 6). These behaviors are commonly observed as zebrafish depression parameters in anxiety models [38]. According to previous reports, long-term high-dose glucocorticoid treatment can clinically increase anxiety, which is known to cause depression or impaired cognitive abilities in patients with sarcopenia [39,40]. These findings suggest that the significant improvement in zebrafish swimming behavior observed with OFBP supplementation may also be correlated with the amelioration of depression-like behaviors induced by Dex-induced muscle atrophy.

In summary, this study demonstrated an excellent improvement in muscle atrophy in zebrafish with OFBP treatment and further highlighted significant enhancement at the myocyte level with its varying concentrations. Long-term Dex treatment is known to induce muscle atrophy in myocytes by increasing the production of proteins that inhibit muscle synthesis, such as myostatin, inducing ROS production associated with mitochondrial dysfunction, and inducing apoptosis which are typical phenomena that occur during age-induced sarcopenia [41,42]. The concentration-dependent treatment of OFBP showed a significant reduction in ROS production in myotubes treated with Dex and alkyl radicals, respectively (Figure 7I,J). This suggests that the improvement effect of OFBP in the Dex-induced muscle atrophy model may be related to the improvement of mitochondrial function. Treatment with OFBP showed a significant cell proliferation effect in myotubes (Figure 4C). These results also indicate that OFBP may have sufficient protective effects against ROS production and subsequent cell death (Figure 7J). Therefore, the industrial application of OFBP, validated through multiple procedures, is expected to promote the eco-friendly utilization of seafood as well as offer a potential application as a high-quality protein food ingredient derived from seafood. Consequently, it is anticipated to be used as a functional food to improve muscle function and support overall health. However, this study did not elucidate the physicochemical properties required for the industrial application of OFBPs, including the constituent amino acids and specific marker components. Additionally, specific evidence regarding the mechanism of muscle function enhancement was not elucidated. Subsequent research to address these limitations is underway and involves the design of extraction processes for mass production and the verification of the anti-sarcopenia effect as a putative functional food ingredient candidate.

## 5. Conclusions

In conclusion, this study focused on OFB and its potential as a valuable resource for the food industry, particularly in addressing muscle-wasting conditions. OFB was found to be rich in protein, making it a highly nutritious food source. The enzymatic hydrolysis of OFB yielded various extracts, with one standout candidate, OFBP demonstrating exceptional protein content and yield. The low-molecularized OFBEs were confirmed via SDS-PAGE and HPLC-MS analysis, with OFBP exhibiting notably low molecular weight, making it a promising candidate for further study. Muscle regenerative potential was investigated using C2C12 myoblasts, revealing that OFBP, along with other OFBEs, promoted glucose metabolism, cell proliferation, and myoblast differentiation. The protective effect of OFBP against Dex-induced myotube atrophy was demonstrated in both in vitro and in vivo, showing significant improvements in muscle function. Furthermore, OFBP significantly exhibited the potential to reduce ROS production in muscle atrophy conditions by Dex stimulation. The study highlights the industrial application potential of OFBP as a high-quality protein food ingredient derived from seafood, with the aim of supporting muscle function and overall health. Future research will address physicochemical properties, constituent amino acids, and specific mechanisms behind these positive effects, paving the way for potential functional food applications.

## Figures and Tables

**Figure 1 antioxidants-12-01755-f001:**
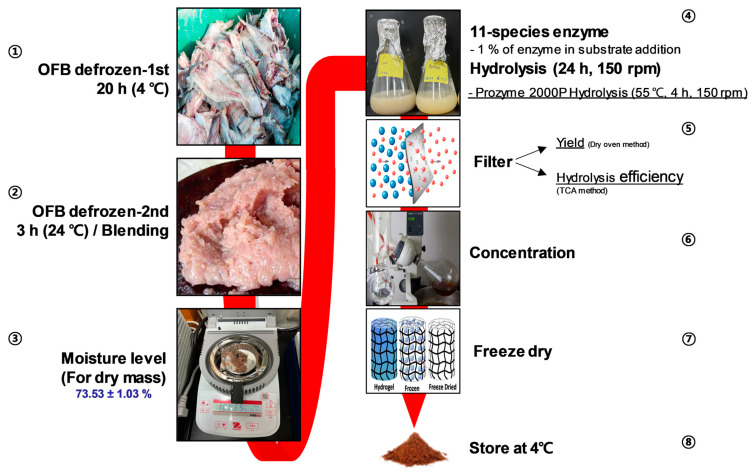
The enzymatic hydrolysis procedure using OFB. During the hydrolysis procedure, steps ①–② involve preparing the raw OFB ingredient for extraction. Step ③ involves calibrating the dry mass in OFB. Step ④ involves setting the enzymatic hydrolysis conditions and inactivating the enzyme. Step ⑤ involves filtering the hydrolysates and confirming the extract performance. Steps ⑥–⑧ involve post-extraction manipulations for storage.

**Figure 2 antioxidants-12-01755-f002:**
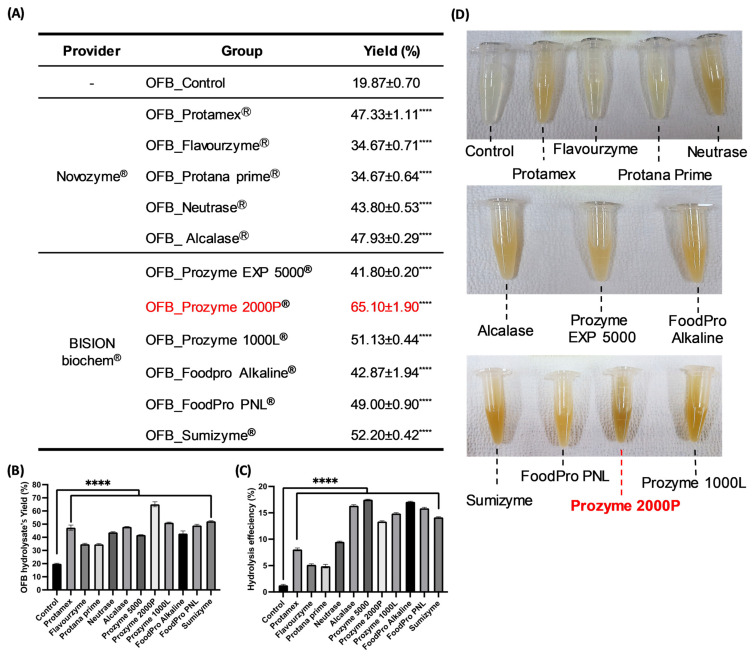
The extract performances of OFBEs. The list of extract yield percentage of OFBEs (**A**) and a graphical figure of hydrolysate yield percentage (**B**). The graphical figure of hydrolysis efficiency percentage (**C**). The comparative images of OFBEs’ turbidity (**D**). Data are expressed as mean ± SD. **** *p* < 0.0001 vs. OFB_Control (n = 3).

**Figure 3 antioxidants-12-01755-f003:**
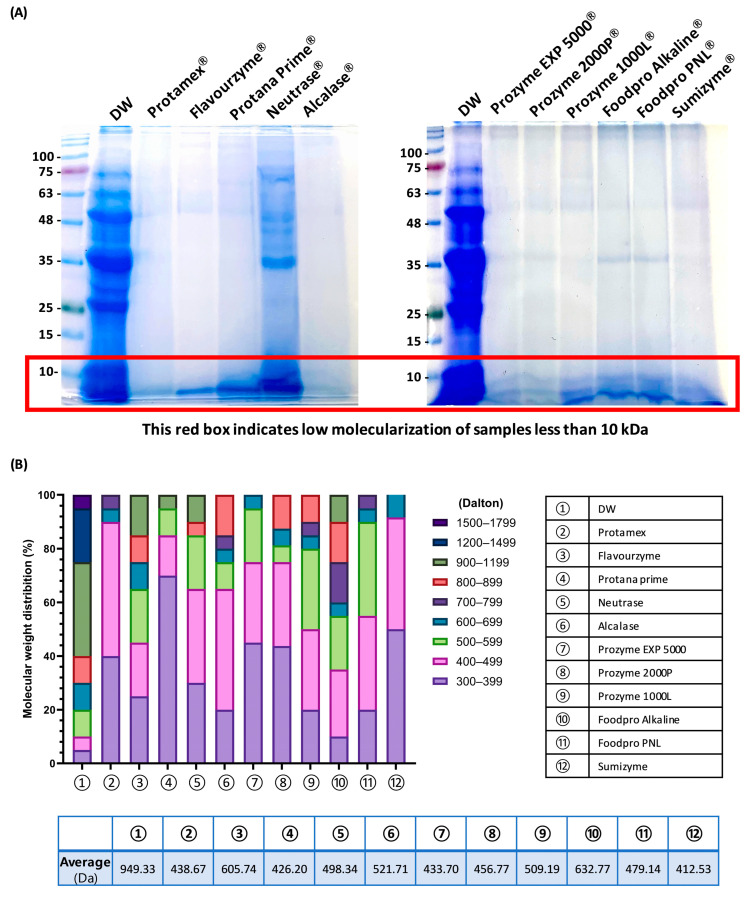
The molecular weight distribution of OFBEs. The SDS-PAGE results indicate the molecular weight distribution of OFBEs stained by Coomassie blue staining (**A**). The percentage of molecular weight distribution of OFBEs was determined by way of the major mass list analysis using HPLC-MS (**B**).

**Figure 4 antioxidants-12-01755-f004:**
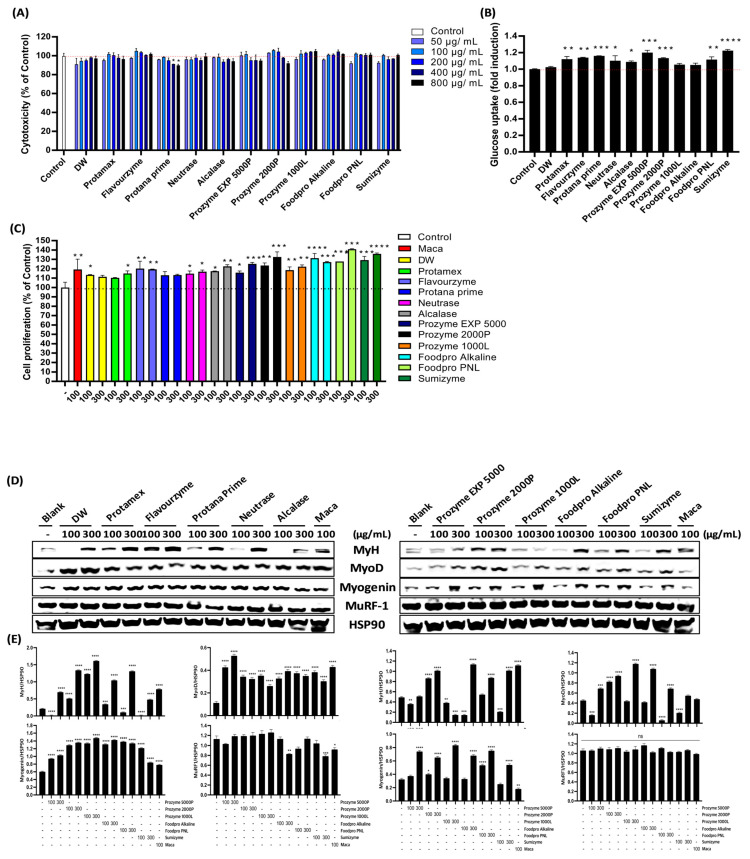
The comparative profiling associating myoblast functional enhancement by OFBEs. The OFBEs treatment induced an extent of cytotoxicity rate (**A**), glucose uptake (**B**), and the regulation of cell proliferation (**C**). The OFBEs treatment-mediated myogenesis-related factors (MRFs) were investigated via immunoblotting (**D**) and the quantified value of an immunoblot was represented graphically (**E**). All the data are shown as mean ± SD (n = 3). Control (Con) is a vehicle-treated group. * *p* < 0.05, ** *p* < 0.01, *** *p* < 0.001, and **** *p* < 0.0001 vs. OFB_Control (n = 3).

**Figure 5 antioxidants-12-01755-f005:**
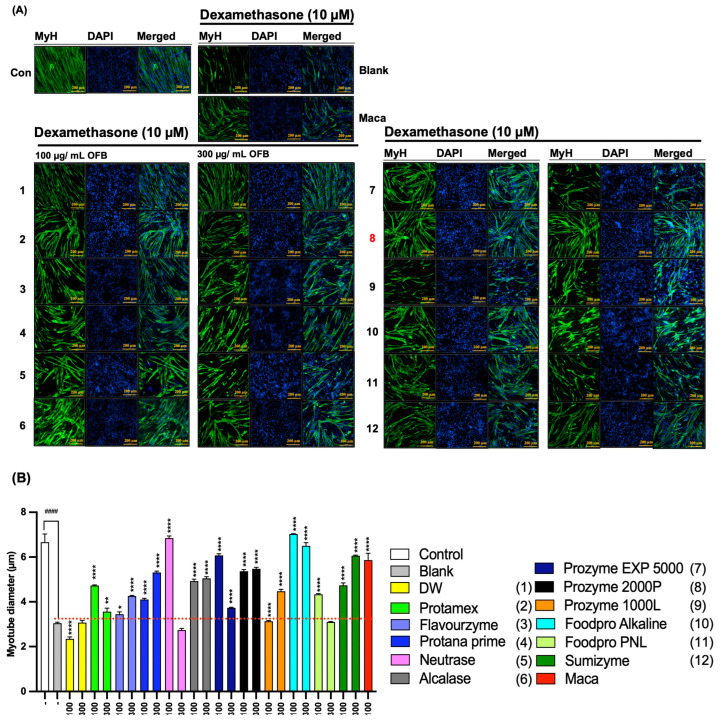
The recovery of myotube formation, which is inhaled by Dex treatment by OFBEs treatment. The cellular Myosin heavy chain (MyH) was targeted for immunofluorescence staining to determine the myotube formation in the Dex-induced C2C12 myotube atrophy model (**A**). The graphically quantified value of MyH immunofluorescent images applied OFBEs (**B**). Entire data are presented as mean ± SD (n = 3). (1) DW, (2) Protamex, (3) Flavourzyme, (4) Protana prime, (5) Neutrase, (6) Alcalase, (7) Prozyme EXP 5000, (8) Prozyme 2000P, (9) Prozyme 1000L, (10) Foodpro Alkaline, (11) Foodpro PNL, and (12) Sumizyme. * *p* < 0.05, ** *p* < 0.01, and **** *p* < 0.0001 vs. Blank. ^####^
*p* < 0.0001 vs. Control. (Sacle bar = 1000 μm).

**Figure 6 antioxidants-12-01755-f006:**
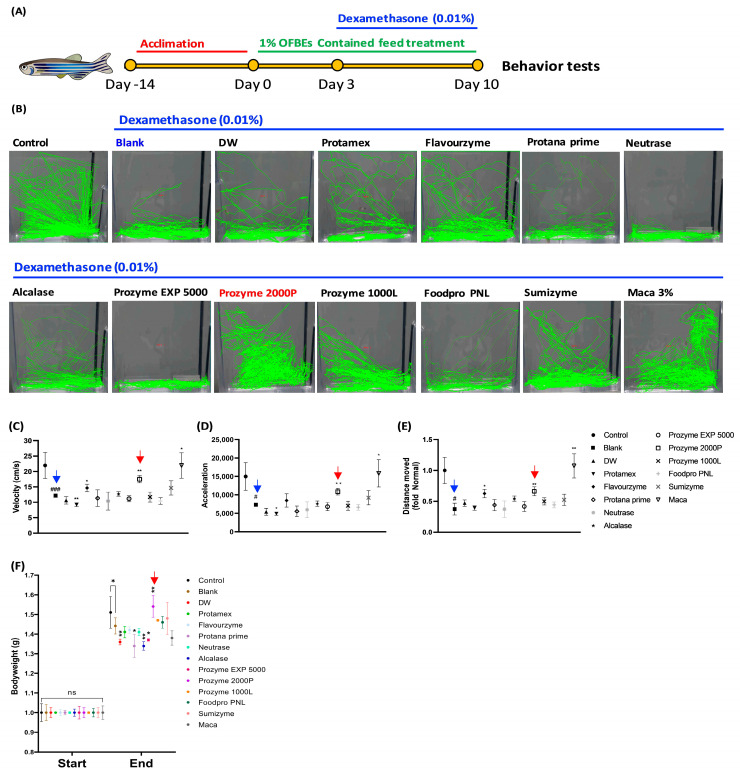
The restoration of imbalance on swimming in Dex-induced muscle atrophy zebrafish model by OFBEs supplementation. The experimental schedule of in vivo model (**A**); 1% of OFBEs contained zebrafish diet was supplemented to zebrafish prior to 0.01% Dex-induced muscle atrophy stimulation. The locomotion tracking of OFBEs supplemented the Dex-induced zebrafish muscle atrophy model (**B**). The graphically quantified values of velocity (**C**), acceleration (**D**), and distance moved (**E**). The change in body weight of the zebrafish experimental group supplemented with OFBEs was measured at the initial and final time points (**F**). All data were indicated to mean ± SD (n = 6). * *p* < 0.05 and ** *p* < 0.01 vs. Blank. ^#^
*p* < 0.05 and ^###^
*p* < 0.001 vs. Control. Blue and red arrows are indicating Dex and OFBP group respectively. ns (Not significant).

**Figure 7 antioxidants-12-01755-f007:**
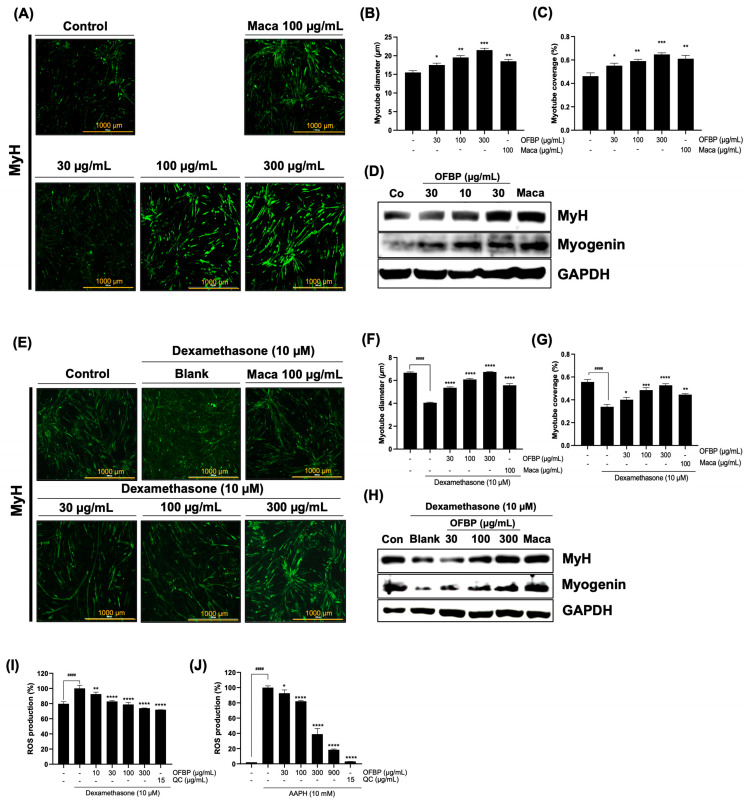
The potential OFBP effect on muscular regeneration in C2C12 myoblast in vitro. The MyH immunofluorescent staining evaluating myotube formation efficiency by way of a series of OFBP concentrations (**A**) and the myotube diameter changes represented by quantitative graphical values of diameter changes in 40 myofibers (**B**), the percentage of myotube coverage change (**C**), and the promotion of MRFs protein expression change through OFBP supplementation in a dose-dependent manner (**D**). The MyH immunofluorescent staining estimating a Dex-induced myotube atrophy restoration by OFBP in dose-dependent treatment in vitro (**E**) and the myotube diameter changes represented by quantitative graphical values of diameter changes in 40 myofibers (**F**), the percentage of myotube coverage change (**G**), and the OFBP-mediated recoveries of MRFs protein expressions, which were inhibited by Dex stimulation (**H**). OFBP reduced the ROS production in Dex-induced C2C12 myotube (**I**) and in AAPH alkyl radical stimulated C2C12 myotube (**J**). All data are expressed as mean ± SD (n = 6). * *p* < 0.05 and ** *p* < 0.01, *** *p* < 0.001, and **** *p* < 0.0001 vs. Blank. ^####^
*p* < 0.0001 vs. Control. (Sacle bar = 1000 μm).

**Table 1 antioxidants-12-01755-t001:** The nutrition constituents of OFB. OFB includes bone, skin, and head that are derived from OF. Data are represented as mean ± SD (n = 3).

Sample	Moisture (%)	Carbohydrate (%)	Protein (%)	Lipid (%)	Ash (%)
OFB (Bone, Skin, Head)	73.53 ± 1.03	3.98 ± 1.01	56.51 ± 1.16	20.37 ± 0.32	18.13 ± 1.65

**Table 2 antioxidants-12-01755-t002:** The list of commercial protease enzymes. Number 1–5 and 6–11 commercial enzymes were purchased from Novozyme LTD and BISION-Biochem, respectively.

No.	Protein Hydrolysis Enzyme	Available Strengths (Range)	Hydrolysis Action	Generation of Peptides or Single Amino Acids	Working pH Range	Working Temperature Range (°C)	Quality Grade
1	Protamex^®^	1.5 AU-A/g	Endo-peptidase	Peptides	6.0–9.0	30–65	Food
2	Flavourzyme^®^	500–100 LAPU/g	Endo-/Exo-peptidase blend	Peptides and Amino acids	4.0–8.0	30–65	Food
3	Protana^®^ Prime	1067 LAPU/g 979 CPDU(A)/g	Leucyl Exo-peptidase	Small peptides and amino acids	3.0–7.0	20–55	Food
4	Neutrase^®^	0.8–1.5 AU-N/g	Endo-peptidase	Peptides	6.0–9.0	30–35	Food
5	Alcalase^®^	2.4–4.0 AU-A/g	Serine endo-peptidase	Peptides	6.5–10.0	60–75	Food
6	Prozyme EXP 5000^®^	5000 LAPU/g	Leucyl exo-peptidase	Small peptides and amino acids	6.0–12.0	55–75	Food
7	Prozyme 2000P^®^	2000 LAPU/g	Leucyl exo-peptidase	Small peptides and amino acids	6.0–9.0	50–60	Food
8	Prozyme 1000L^®^	200,000 PC/g	Endo-peptidase	Peptides	7.0–8.0	50–55	Food
9	FoodPro PNL^®^	1600 AZO/g	Endo-peptidase	Peptides	6.0–8.0	50–55	Food
10	FoodPro Alkaline^®^	580,000 DU/g	Alkaline Serine Endo-peptidase	Peptides	8.5–9.0	60–70	Food
11	Sumizyme DPP-G^®^	100 U/g	Dipeptidyl-aminopeptidase	Peptides	7.0–7.5	55–65	Food

Activity Unit (AU), Leucyl AminoPeptidase Unit (LAPU), Protease Unit on L-tyrosine basis (PC), and Dextrinizing Unit/g (DU).

**Table 3 antioxidants-12-01755-t003:** The analysis of nutritional constituents of OFBEs. All the data presenting carbohydrate, protein, lipid, and ash are displayed as mean ± SD. * *p* < 0.05, ** *p* < 0.01, *** *p* < 0.001, and **** *p* < 0.0001 vs. OFB_Control (n = 3).

Provider	Samples	General Composition (%)			
		Carbohydrate	Protein	Lipid	Ash
-	OFB_Control (DW)	0.66 ± 0.11	93.83 ± 2.84	0.69 ± 2.37	4.82 ± 0.40
Novozyme^®^	OFB_Protamex^®^	7.27 ± 0.25 ****	63.56 ± 0.95 **	25.64 ± 0.98 *	3.53 ± 0.19 *
	OFB_Flavourzyme^®^	7.43 ± 0.42 ****	56.00 ± 0.95 **	32.82 ± 1.07 **	3.75 ± 0.15 *
	OFB_Protana prime^®^	12.72 ± 0.19 ****	47.48 ± 1.89 **	36.33 ± 1.57 **	3.47 ± 0.14 *
	OFB_Neutrase^®^	3.26 ± 0.50 **	65.46 ± 2.84 *	28.37 ± 3.10 *	2.91 ± 1.05
	OFB_Alcalase^®^	11.60 ± 0.31 ****	68.29 ± 1.89 *	14.56 ± 2.15 *	5.55 ± 0.84
BISION-Biochem^®^	OFB_Prozyme 5000P^®^	8.15 ± 0.00 ****	77.75 ± 0.00 *	9.56 ± 0.42	4.54 ± 0.59
	OFB_Prozyme 2000P^®^	4.20 ± 0.19 ****	85.72 ± 6.62	6.38 ± 4.99	3.70 ± 0.25
	OFB_Prozyme 1000L^®^	9.78 ± 0.15 ****	57.89 ± 0.95 **	29.25 ± 0.82 **	3.08 ± 0.06 **
	OFB_Foodpro Alkaline^®^	9.03 ± 0.13 ****	62.62 ± 1.89 *	24.76 ± 1.51 *	3.59 ± 0.11 *
	OFB_FoodPro PNL^®^	0.63 ± 0.13	62.62 ± 7.57	33.61 ± 5.54 *	3.14 ± 0.14 **
	OFB_Sumizyme^®^	1.67 ± 0.08 ***	58.83 ± 1.89 **	36.53 ± 1.54 **	2.97 ± 0.21 **

## Data Availability

Data is unavailable due to ethical restriction.

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
