# Peer review of "Exploring the Potential of Olive Flounder Processing By-Products as a Source of Functional Ingredients for Muscle Enhancement"

_antioxidants, 2023, doi:10.3390/antiox12091755_

Round 1

Reviewer 1 Report

Summary

This manuscript reports the anti-muscular atrophic effect of enzymatic hydrolysis of olive flounder by-products (OFBEs) on C2C12 murine myoblasts and zebrafish. The OFBE produced by Prozyme 2000P contained higher protein composition compared to the control. This OFBE promoted cell proliferation and myogenic differentiation of C2C12 cells, and showed anti-atrophic effect against dexamethasone (Dex)-induced atrophy on C2C12 myotubes, in part, by inhibiting ROS production. In zebrafish, this OFBE reversed Dex-exacerbated locomotion. In conclusion, OFBE by Prozyme 200P potentially recovers skeletal muscle atrophy. Almost experiments have been adequately performed and described. Revising some points below will make this study perfect for publication in Antioxidants.

Comments

1.          Line 137: “1*10^5 cells/well”: What type of well plate did the authors use?

2.          Tables 1 and 2: The number of experiments should be described.

3.          Figure 5A: [1] The aspect ratio of the images seems to be in inappropriate (vertically compressed). [2] Scarle bar should be displayed. (3) There is no explanation for the numbers (1~12) displayed on the left side of the images.

4.          Figure 6F: The body weights of the zebrafish (n = 6) must be indicated as the mean ± SD of the absolute values (g or mg) and analyzed statistically. Body weights were increased in the Prozyme 2000P group but also in the control group. Is there a significant difference between them? Did other OFBEs decrease body weight compared with to control group? If so, why? The positive control, Maca, did not work? These results need to be accurately analyzed, described, and discussed.

5.          Figures 7A and 7E: Scale bars are displayed but their scales are not described. This should be in the legend.

6.          The function and mechanism of anti-atrophy effect of Maca need to be briefly introduced for the readers who are not familiar in this field.

Minor points

7.          Line 56: “PUFA” should be defined.

8.          Lines 123, 271, 313, 352, 382, and 415: “Hyun et al., (2023).” should be properly cited. I think it’s inserted by mistake. Check the manuscript carefully.

9.          Line 129, 152: “phosphate-buffered saline (PBS)” should be “PBS”.

10.          Line 535: The reference number should be “5”.

Author Response

Thank you for your kind words. We have carefully reviewed and incorporated your advice, so please check. 

Reviewer 2 Report

In this manuscript, the authors have tested the beneficial effect of olive flounder byproduct after enzymatic treatment on myotube differentiation and muscle atrophy. The study is interesting. The manuscript is well written.  However, there are several demerits in the study which needs to be addressed before its final acceptance for publication. My comments are provided below

1. The authors have used a crude extract in their study. The enzymes were also not separated from the OFBE. It is not clear which components of the OFBE is responsible for the function. The authors should discuss this issue.

2.  In Figure 4, the study has been performed with only one cell line - C2C12. The authors didnot mention whether control is vehicle treated or enzyme only treated. The authors should show the result of MTT assay for cytotoxic assay. 

3. In Fig 5, the authors should show the effect of Dex on myoyubes by staining for AChR by BTX and western blot analysis of myogenic markers.

4. In fig 6, the authors should first demonstrate the intake of  OFBE in zebrafish model.

5. In fig 7, how many myotubes were analyzed for diameter measurement study. The authors can use the primary myoblast cells to strengthen the claim.

There are minor spelling mistakes in the text.

Author Response

(The authors gave the same response as above.)

Round 2

Reviewer 1 Report

The authors have revised the manuscript according to the reviewer's comments.

Reviewer 2 Report

The authors have addressed all my concerns in the revised manuscript. I support the publication of manuscript.

Minor spelling mistakes in the text.